# Replicate Studies Separated by 40 Years Reveal Changes in the Altitudinal Stratification of Montane Passalid Beetle Species (Passalidae) in Mesoamerica

Cristian Fernando Beza-Beza [1,2,*,†] , Camilo Rivera [3,4,†], Diego Pons [5] , Duane McKenna [1] and Jack C. Schuster [4]

1   Department of Biological Sciences, Center for Biodiversity Research, University of Memphis, Memphis, TN 38152, USA
2   Department of Entomology and Plant Pathology, North Carolina State University, Raleigh, NC 27695, USA
3   Department of Evolutionary Entomology, University of Neuchatel, 2000 Neuchatel, Switzerland
4   Laboratorio de Entomología Sistemática, Universidad del Valle de Guatemala, Guatemala City 01015, Guatemala
5   Department of Geography and the Environment, University of Denver, Denver, CO 80210, USA
*   Correspondence: cfbezabe@ncsu.edu; Tel.: +1-919-515-3519
†   These authors contributed equally to this work.

**Abstract:** Two patterns are apparent in the altitudinal distribution of Neotropical passalid beetles: (a) species that occur only in lowland forest habitats but have broad geographic distributions, and (b) montane endemic species with relatively limited distributions. The transition zone between these distributions in upper Mesoamerica occurs, on average, at approximately 1500 m above sea level (a.s.l.). We studied the altitudinal stratification of passalid beetle communities living on two volcanoes in Guatemala (Atitlan and Santa Maria), revisiting a study conducted in 1981 by MacVean and Schuster. We collected passalid beetles at the same study sites and compared the community composition along the altitudinal gradient. We collected all but one of the species reported by MacVean and Schuster and found three additional species. We observed two key differences in the passalid communities observed in 1981 versus the present: (a) for the Atitlan site, the species' turnover line from lowland to montane species shifted from 1600 to 1800 m a.s.l.; and (b) in both volcanoes, we collected passalid beetles well above 2700 m a.s.l., which was the upper limit at which they were found in 1981. Both observations are consistent with a shift of the passalid beetle community to higher elevations, perhaps in response to changes in local climate/habitat conditions, including increased temperatures and changes in forest composition.

**Keywords:** altitudinal gradients; endemism; Atitlan volcano; Santa Maria volcano; Mesoamerican humid montane forest





## 1. Introduction

Insect populations worldwide are declining at an alarming pace, which has been deemed an "apocalypse" [1]. Among the contributors to the decline in insect biodiversity are the loss of habitat and human-induced climate change. Tropical high-altitude biota are particularly vulnerable to these changes [2,3] due to the reduction of their habitat, which is already naturally scarce. It is expected that species ranges in tropical montane habitats suffer range contractions and likely shifts to higher altitudes as an adaptive measure [3,4]. One of the tools used to monitor the response of these species is the historical measurement and comparison of their altitudinal distribution ranges. Re-survey studies have shed some light on the responses of insects to climate change and altitudinal range shifts. The evidence shows that the responses of different insects to climate change have been variable; some species have shown altitudinal shifts upward, whereas some have shifted downward, and others show no changes in their ranges [5–8]. However, since most of these studies have been conducted in temperate areas,

there is a huge need for similar data to be collected in the tropics, where most of the insect species exist [3,7]. Here, we take advantage of one detailed historical study of the altitudinal distributions of Passalid beetles in Guatemala [9] and a restudy to shed light on what is happening to montane endemic species in the Neotropics.

Passalid beetles are relatively large saproxylophagous beetles living and feeding inside rotting wood. In the Americas, Passalidae can be found from sea level to approximately 3000 m above sea level (a.s.l.), with their highest species richness occurring between 1000 and 1500 m a.s.l. [10]. Passalid beetles have served as a model system for establishing conservation areas and play an essential ecological role in wood decomposition [11,12]. The geographic distribution of species within the family Passalidae is highly correlated with altitude [9,13], with broadly distributed fauna in the lowlands and narrowly distributed locally endemic fauna occurring at higher altitudes. About 80% of the species diversity of passalid beetles in Guatemala is associated with montane ecosystems.

Mid-to-high altitudes in Mesoamerica host what is known as Humid Montane Forest (HMF), this is naturally patchy and characterized by a high degree of endemism [14–16]. The HMF is one of the most threatened ecosystems in Mesoamerica [14]. The volcanoes in the Guatemalan Southern Volcanic Chain (GSVC) are an area of endemism for beetles in the family Passalidae that are restricted to high altitudes [16,17]; there are nearly 90 extant species of Passalidae in Guatemala of which at least 33 occur in the GSVC [18] (personal observation). In the GSVC, much of the forest at mid elevations has been cleared for agricultural use (mainly coffee plantations) [9], it faces the additional threat of a reduction in area due to rising temperatures resulting from climate change and an altitudinal shift of coffee plantations to accommodate these temperature changes [19,20]. Organisms with similar distributions and endemism patterns in Mesoamerica such as salamanders experienced a reduction in species diversity at high altitudes between 1975 and 2005, primarily due to changes in microclimate in the habitats where they occurred [21].

In 1981, MacVean and Schuster published a study of the distribution and biogeographic history of passalid beetles in the GSVC. The study included a species list with elevational ranges for species found along an altitudinal gradient on seven volcanoes. Systematic transects were performed for two volcanoes (Atitlan and Santa Maria) and data from the other five volcanoes were gathered from sparse collection events. The passalid beetle fauna was uniform and species had similar elevational ranges among the volcanoes in the GSVC [9]. The authors identified two types of fauna: one in low-to-mid elevations (from 800 to 1500 m a.s.l.) and a strictly montane one above 1500 m a.s.l. Despite extensive collection efforts (25 logs were inspected on each volcano), no passalid beetles were collected above 2700 m a.s.l. The low-to-mid-elevation fauna comprised species with broad geographic distributions (e.g., *Passalus punctiger*, LePeletier and Serville, 1825; *Verres hageni*, Kaup, 1871), and the high-elevation fauna included mainly montane endemic species (e.g., *Ogyges laevissimus*, Kaup, 1868; *Pseudacanthus junctistriatus*, Kuwert, 1891; and *Pseudacanthus subopacus*, Bates, 1886). We revisited the two elevational transects (Atitlan and Santa Maria) established by MacVean and Schuster [9] to compare the current community assembly and elevational stratification of passalid beetles in the GSVC with those of 40 years ago. We hope that the results from this research can be used to inform conservation efforts in the region.

## 2. Materials and Methods

### 2.1. Study Site and Data Collection

The Guatemalan Southern Volcanic Chain (GSVC) consists of 15 volcanoes, all of which are part of the Chortís Volcanic Front Province [22]. These volcanoes are located behind the Guatemala Coastal Plain in front of the Pacific Ocean (Figure 1). They occur in clusters flanked by silicic calderas (e.g., Santa Maria + Santiaguito + Santo Tomas, Cerro Quemado, and San Pedro + Atitlan + Tolimán clusters) [22]. Because of their proximity to the Pacific coast of Mesoamerica, more rain falls on the southern sides of the volcanoes

than on other sides, causing a rain-shadow effect and resulting in more humid forests along the southern slopes compared to the northern slopes.

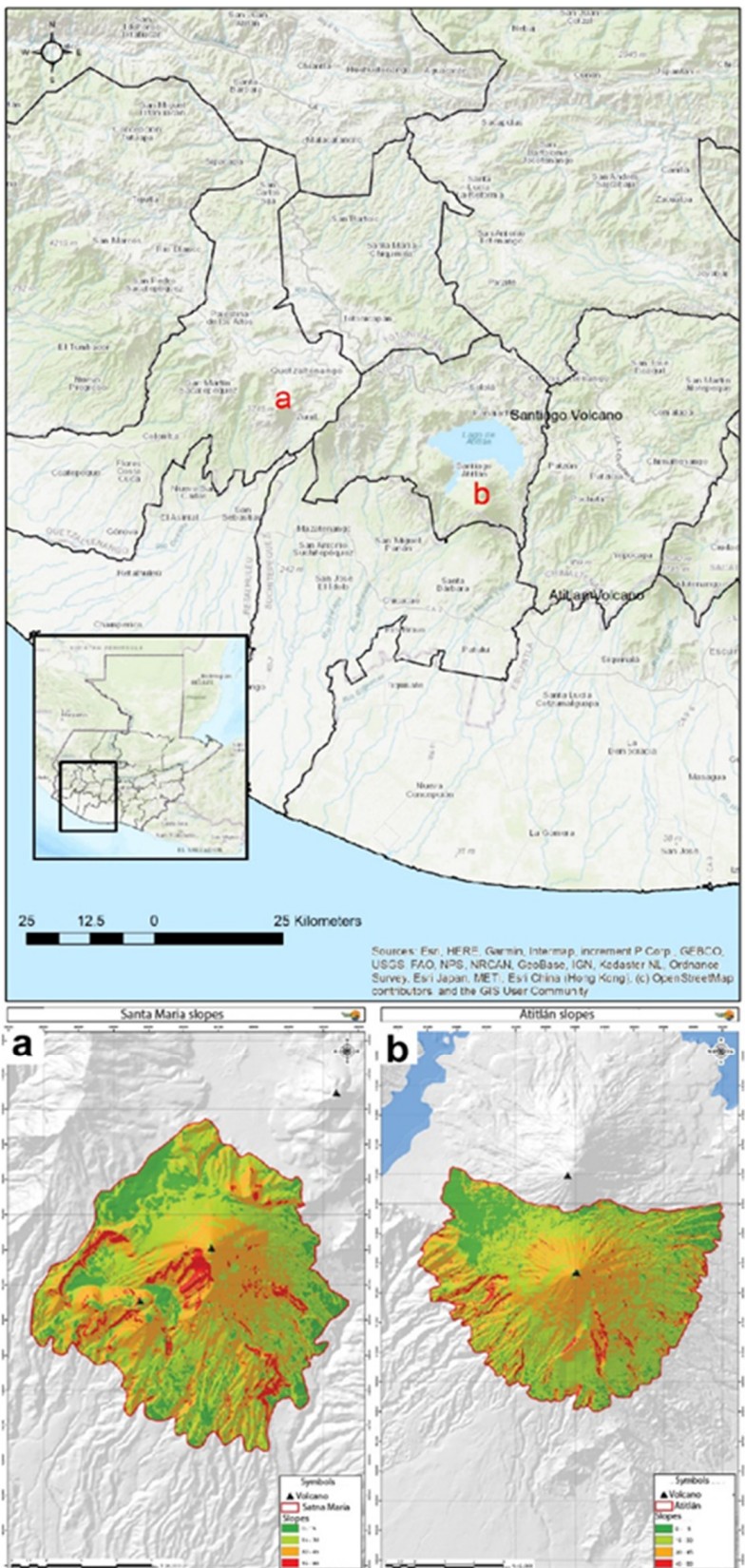

**Figure 1.** Map of the Southern Guatemalan Volcanic Chain and slope profile for the (**a**) Santa Maria and (**b**) Atitlan volcanoes.

Atitlan and Santa Maria are stratovolcanoes with altitudes of 3535 m and 3745 m, respectively [23]. The vegetation of Atitlan corresponds to "low subtropical moist forest" at lower elevations and "lower montane subtropical moist forest" at higher elevations, following Holdridge's [24] life zones. On Santa Maria, the northern slopes are forested, whereas the upper southern slope is unvegetated. The vegetation in Santa Maria comprises a "lower montane very moist subtropical forest" [25]. On the southern slope of Santa Maria is a massive lava dome known as Volcan Santiaguito [23], which is still active and lacks vegetation. The vegetation at lower altitudes on both volcanoes has been replaced with farmland, mainly coffee plantations [9].

We collected passalid beetles along the elevational gradients on the Atitlan and Santa Maria volcanoes. In doing so, we attempted to replicate to the best of our ability the transects described in MacVean and Schuster [9]. At Atitlan, we performed an elevational transect along the southern slope. At Santa Maria, we performed one transect between 800 and 2400 m a.s.l. on the southern slope and another between 2400 and 3600 m a.s.l. on the northern slope. The split of the Santa Maria transect into two is because the Santiaguito lava dome on the southern slope of Santa Maria is unvegetated and unsuitable as a habitat for passalid beetles; this was also done for the study of MacVean and Schuster [9].

Without abundance or probability occurrence data for the species in the area, it is difficult to know if the observed differences are due to ecological changes, substrate availability (rotting logs), or collection efforts. Measuring the collection efforts for bess beetles is difficult for multiple reasons. For example, we still do not understand if collection efforts are better standardized by measuring the time spent searching an area, the number of logs, wood volume, or covered area. Given their sub-social nature, the number of individuals is not an effective measure of abundance. To ensure that both sampling periods are comparable, we sampled beetles in rotting logs at approximately every 200 m of altitudinal change on both volcanoes (Figures S1 and S2). The geographical coordinates for the collection points are detailed in File S1. We searched for suitable rotting wood for passalids in each elevational belt, including stumps, twigs, and logs, as described in McVean and Schuster [9]. Sampling efforts at each collection spot were dependent on log availability, similar to [9]; this number varied between each collection site. Beetles were collected and brought to the Colección de Artrópodos de la Universidad del Valle de Guatemala for identification. We identified the specimens based on the "key to the American genera of Passalidae" [26] and reliable identifications by Jack C. Schuster. To reflect the sampling splits, we recorded the elevational range for each species. We tabulated the data in a presence–absence matrix with each volcano as a unit (volcano level) and in 200 m elevational belts for 1981 and 2017/2018 (File S2).

*2.2. Faunistic Analysis*

We characterized each volcano and compared the sampling periods by the species composition of each elevational belt (200 m of altitude each). To determine if the elevational belts would form clusters based on shared species, we performed Hierarchical Cluster Analysis (HCA) for each volcano and each sampling period using the base function hclust in R [27]. The distance matrix for performing HCA was obtained with the function vegdist in the R package vegan [28] using methods = "jaccard". We limited the HCA to the altitudinal belts where passalid beetles were collected in both collection periods; between 1200 and 2700 m a.s.l. for Atitlan and between 1000 and 2700 m a.s.l. for Santa Maria. To select the number of clusters within each HCA, we identified a cut point that produced a reasonable number of clusters, and then we checked the biological significance of those clusters by comparing the species composition of each cluster member. Species turnover lines were defined as the altitude where one cluster shifts to another.

*2.3. Change in Climatic Variable*

To examine the climate change trends for the area, we used three key climatic variables, temperature, precipitation, and potential evapotranspiration. We used the data sets from

Harris et al. [29] for temperature and potential evapotranspiration. For precipitation, we used the data sets from Funk et al. [30]. To obtain tendency lines, we restricted the data to values from the Guatemalan rainy season (May to October). We averaged the values for the rainy season; for temperature, the values are the average daily temperatures in degrees Celsius. For precipitation and potential evapotranspiration, the values are the average daily mm. Additionally, we performed a simple linear regression in R v 4.0.4 [27] using R Studio v 1.4.1106 [31]. Finally, to check the significance of the correlation slope, we performed a *t*-test for linear regression.

## 3. Results

### 3.1. Atitlan Volcano

We found passalid beetles between 800 and 2900 m a.s.l. We recorded a total of 10 species (Table 1). We could not assess an elevational range for *Spurius bicornis* (Truqui, 1857), which we collected at only one elevation. Likewise, the elevational range of *Oileus sargi* (Kaup, 1871) (1700–1750 m a.s.l.) and *O. laevissimus* (2300–2350 m a.s.l.) consisted of 50 m (Table 1). The highest species counts corresponded to the first three elevational belts, with four species in each belt. The lowest species count (a single species) was in the highest elevational belt (2600–2799 m a.s.l.).

**Table 1.** Altitudinal ranges (m a.s.l.) of passalid beetles reported in MacVean and Schuster (1981) [9] compared with the beetles collected in this study.

| | Atitlan | | Santa Maria | |
|---|---|---|---|---|
| | **1981** | **Current** | **1981** | **Current** |
| *Paxillus leachi* | – | 850 | – | – |
| *Passalus punctiger* | – | – | 800 | 1016 |
| *Rhodocanthopus caelatus* | 1500 | 850–1668 | 1150 | – |
| *Passalus punctatostriatus* | – | 1100–1300 | 1150–1550 | – |
| *Odontotaenius striatopunctatus* | 1400 | 850–1300 | 1150 | 1230–1377 |
| *Verres hageni* | 1400 | 850 | 1550 | 1043 |
| *Spurius bicornis* | 1350–1500 | 1100–1668 | 1550 | – |
| *Oileus sargi* | – | 1700–1750 | 1550 | 1423–2274 |
| *Arrox agassizi* | – | – | – | 2214 |
| *Chondrocephalus debilis* | 1550–2500 | – | 1750 | – |
| *Chondrocephalus purulensis* | 1950–2300 | 2049–2300 | 1750–2300 | – |
| *Ogyges laevissimus* | 2500 | 2300–2350 | 2100–2600 | – |
| *Chondrocephalus granulifrons* | 2500–2700 | 1900–2900 | 2200–2650 | 2700–3300 |
| *Pseudacanthus subopacus* | – | – | 2300 | 1814 |
| *Vindex c.f. sculptilis* | 2500 | – | – | 1978 |
| *Pseudacanthus junctistriatus* | – | – | – | 2395–3124 |
| *Proculini* sp. | – | – | – | 3498 |
| Total # Species | 9 | 10 | 12 | 10 |

For the 1981 data, the HCA recovered three clusters within Atitlan (Figure 2A). One comprised all altitudes below 1599 m a.s.l. (cluster 1 Figure 2A), the second constituted the altitudinal belts between 1600 and 2399 m a.s.l. (cluster 2 Figure 2A), and the third cluster contained the altitudinal belts between 2400 and 2799 m a.s.l. (cluster 3 Figure 2A). Each cluster comprised a distinct species community (Table 2) marked by a nearly complete species turnover. For this collection event, we detected two species turnover lines (1600 and 2400 m a.s.l.) (Figure 2C).

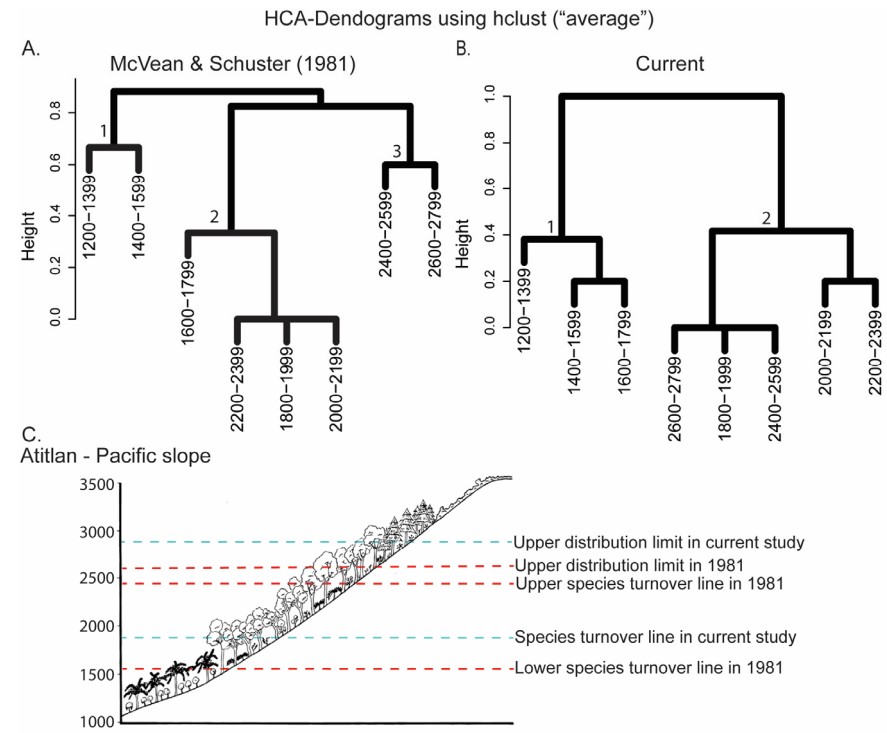

**Figure 2.** (**A**) Dendrogram of the HCA for the presence–absence of passalid beetle species in 200 m altitudinal belts (m a.s.l.) for the Atitlan volcano in 1981. (**B**) Dendrogram of the HCA for the presence–absence of passalid beetle species in 200 m altitudinal belts (m a.s.l.) for the Atitlan volcano in the current study. (**C**) Pacific slope of the Atitlan volcano, modified from McVean and Schuster (1981), showing the species turnover line and upper distribution limit of Passalidae for each sampling period.

**Table 2.** Species composition of the HCA clusters for each volcano and sampling period.

| Volcano | Sampling Period | Cluster # | Species Composition |
|---|---|---|---|
| Atitlan | 1982 | Cluster 1 (1200–1599) | *R. caelatus, C. debilis, Od. striatopunctatus, S. bicornis, V. hageni* |
| | | Cluster 2 (1600–2399) | *C. debilis, C. purulensis* |
| | | Cluster 3 (2400–2799) | *C. debilis, C. granulifrons, O. laevissimus, V.* sp. |
| | Current | Cluster 1 (1200–1799) | *R. caelatus, P. punctatostriatus, O. striatopunctatus, Oi. sargi, S. bicornis* |
| | | Cluster 2 (1800–1799) | *C. granulifrons, C. purulensis, O. laevissimus* |
| Santa Maria | 1982 | Cluster 1 (1200–1599) | *R. caelatus, P. punctatostriatus, Od. striatopunctatus, Oi. sargi, S. bicornis, V. hageni* |
| | | Cluster 2 (1600–2799) | *C. debilis, C. granulifrons, C. purulensis, Ps. subopacus, O. laevissimus* |
| | Current | Cluster 1 (1000–1199) | *P. punctiger, V. hageni* |
| | | Cluster 2 (1200–1399) | *O. striatopunctatus* |
| | | Cluster 1 (1400–2199) | *Oi. sargi, Ps. subopacus, V.* sp |
| | | Cluster 2 (2200–2799) | *Oi. sargi, C. granulifrons, Ps. junctistriatus, A. agassizi* |

The HCA recovered two clusters for the current sampling period (Figure 2B). One comprised all altitudes below 1800 m a.s.l. (cluster 1 Figure 2B), whereas the other contained all altitudes above 1800 m a.s.l. (cluster 2 Figure 2B). The recovered clusters did not share any species. The species composition for each cluster can be seen in Table 2. We detected one species turnover line (1800 m a.s.l.) (Figure 2C) for this collection event.

Cluster 1 (low-mid-elevation cluster) in 1981 and the current collection event shared a similar species composition (Table 2). Cluster 2 in the current data set had a similar species composition to clusters 2 and 3 in the 1981 data set (Table 2). This suggests that the transition between low-elevation fauna (cluster 1 for each collection period) and montane fauna occurred at an elevation range of 1600 to 1800 m a.s.l. (Figure 2C).

### 3.2. Santa Maria Volcano

In Santa Maria, we found logs containing passalid beetles between 1000 and 3500 m a.s.l, collecting ten species (Table 1). We could not assess the elevational range for *P. punctiger*, *V. hageni*, *Ps. subopacus*, *Vindex* sp., *Arrox agassizi* (Kaup, 1871), and an undescribed species of Proculini. The elevational belt with the highest species count (three species) was between 2200 and 2399 m a.s.l.; all other elevational belts contained one or two species. We also recorded an undescribed species of Proculini at 3498 m a.s.l.; this was the only species collected at this elevation.

For the 1981 data, the HCA recovered two clusters within Santa Maria (Figure 3A). One comprised all altitudes below 1600 m a.s.l. (cluster 1 Figure 3A), and the second constituted all altitudinal belts above 1600 m a.s.l. (cluster 2 Figure 3A). The recovered clusters did not share any species. The species composition for each cluster can be seen in Table 2. We detected one species turnover line (1600 m a.s.l.) (Figure 3C) for this collection event.

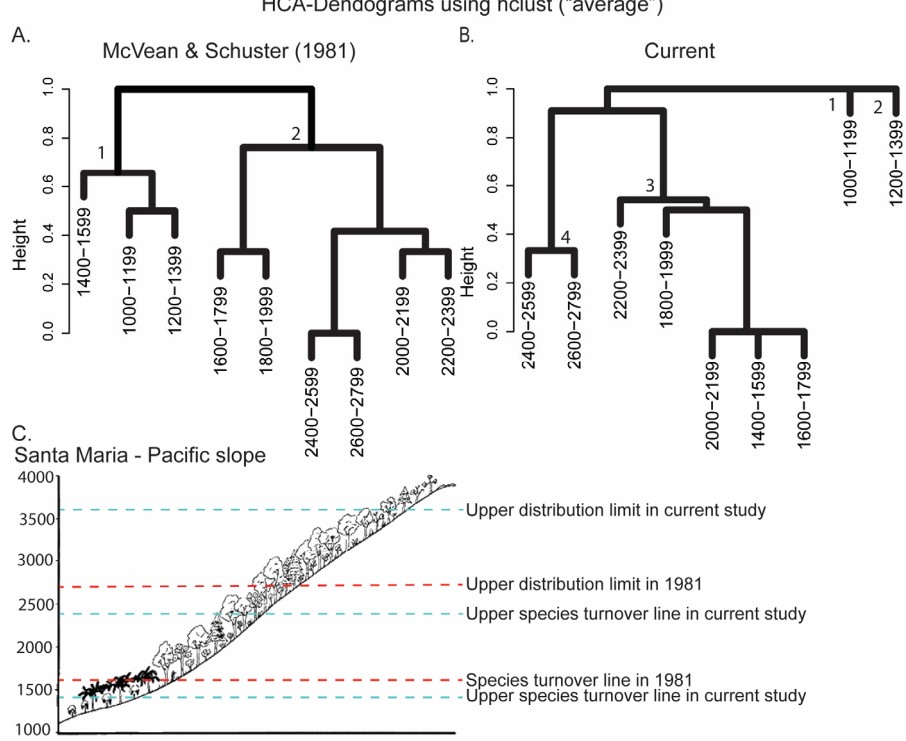

**Figure 3.** (**A**) Dendrogram of the HCA for the presence–absence of passalid beetle species in 200 m altitudinal belts (m a.s.l.) for the Santa Maria volcano in 1981. (**B**) Dendrogram of the HCA for the presence–absence of passalid beetle species in 200 m altitudinal belts (m a.s.l.) for the Santa Maria volcano in the current study. (**C**) Pacific slope of the Santa Maria volcano, modified from McVean and Schuster (1981), showing the species turnover line and upper distribution limit of Passalidae for each sampling period.

For the current data, the HCA recovered four clusters. The first two elevational belts, 1000–1199 m a.s.l. and 1200–1399 m a.s.l., were not part of any other altitudinal belt(s) and each constituted a cluster (clusters 1 and 2 Figure 3B). The other altitudinal belts formed two clusters, one comprising all altitudes between 1400 and 2399 m a.s.l. (cluster 3 Figure 3B) and the other comprising altitudes above 2400 m a.s.l. and 2700 m a.s.l (cluster 4 Figure 3B).

Clusters 3 and 4 (Figure 3B) were the only clusters that shared species (one species shared: *Oi. sargi*). The species composition of each cluster is shown in Table 2. The species composition of cluster 1 in the 1981 data (Figure 3A) was similar to that of the two belts of the current data set that did not cluster with other altitudinal belts (Table 2, Figure 3B). At the same time, cluster 2 of the 1981 (Figure 3B) data set shared species with clusters 3 and 4 of the current data set (Table 2, Figure 3B).

## 4. Discussion

We observed changes in the altitudinal stratification of the Passalid beetle fauna at Atitlan and Santa Maria, the volcanoes only shared 4 of the 16 species reported in the present study. Three of those species (*R. caelatus*, *Od. striatopunctatus*, *V. hageni*) were low-to-mid-elevation species, whereas the only montane species shared was *C. granulifrons*. In comparison, MacVean and Schuster [9] found eight species shared between the two volcanoes. In MacVean and Schuster [9], four of the shared species (*P. punctatostriatus*, *Od. striatopunctatus*, *V. hageni*, *S. bicornis*) were considered low-to-mid-elevation species, and the other four (*C. debilis*, *C. purulensis*, *C. granulifrons*, *O. laevissimus*) were strictly montane species. This is a marked difference, especially considering that we collected similar assemblages for both collection periods (Table 1). We registered almost all 14 species reported by MacVean and Schuster [9], except for *C. debilis*, which was not detected in this study. In addition, we collected *Pa. leachi*, *A. agassizi*, and an undescribed species with uncertain generic placement within Proculini (sp. "*incertae sedis*"). Of these newly recorded species, *Pa. leachi* and *A. agassizi* are widely distributed throughout Mesoamerica and were expected to be present on the volcanoes. These differences indicate that the high-altitude fauna has changed more drastically than the low-to-mid-elevation fauna.

Similar to other studies, we found a high degree of altitudinal specialization in the bess beetle communities of Atitlan and Santa Maria (Figures 2 and 3). Elevational stratification of the bess beetle fauna in Mesoamerica is a relatively well-documented phenomenon (e.g., MacVean and Schuster [9]; Chamé-Vázquez et al. [13]; Serrano-Peraza et al. [32]) and is known to occur in other Neotropical beetles [33,34]. Chamé-Vázquez et al. [13] undertook an altitudinal transect in el Soconusco, Chiapas, México, and recovered a similar pattern of elevational stratification as MacVean and Schuster [9] and the present study. The authors found two faunal assemblages: (a) between 50 and 1500 m a.s.l., most species in the assemblage were characterized by a broad geographical distribution across the New World, and (b) above 1500 m a.s.l., they found montane endemic genera (e.g., *Chondrocephalus* Kuwert, *Pseudacanthus* Kaup, *Undulifer* Kaup, and *Vindex* Kaup).

Given the limitation of this study outlined in the methods, we suggest that alternative approaches (e.g., Royle and Nichols [35]) for estimating abundance values could help compare passalid beetle communities effectively. Royle and Nichols [35] presented a method that allows the estimation of abundance from repeated observations (>2) of presence–absence data. To apply such a method, continuous sampling in the area is necessary. Such an approach would help establish a baseline for the passalid beetle community in the GSVC. Despite these limitations, we observed two main differences in the community composition of Passalidae in the GSVC:

(1)   At the Atitlan volcano, MacVean and Schuster [9] detected a turnover from lowland to montane species at 1500 m a.s.l. (1600 m a.s.l. in our analyses). However, hierarchical cluster analyses of our data indicated that the current species turnover was at 1800 m a.s.l. The 1500 m a.s.l. boundary for species turnover in Passalidae tends to be consistent across altitudinal gradient studies of Passalidae (e.g., Tingo Maria region in Peru [36] and Soconusco México [13], Parque Nacional el Trifinio in El Salvador [32]).

In these locations, species turnover is correlated with a change in vegetation from lower montane forest to humid montane forest such as cloud forest. Mesoamerican forest cover declined from 72% in the 1980s to 42% today and humid montane habitats are one of the most threatened environments [14]. Although assessments of the structural composition of the forest are not available, land-use coverage (e.g., humid montane forest) between 1500 and 1800 m a.s.l. on the south side of the Atitlán volcano has not significantly changed in the last 50 years (Figure 4).

In light of these findings, we should consider the role of other environmental variables, such as increasing mean temperature, unchanged mean precipitation, and an increase in the mean potential evapotranspiration registered in the area, as possible drivers of these changes (Figure 5, File S3). The interactions between these variables could explain the upward shift in the species turnover line in the passalid community. The significant increase in temperatures and its relationship with precipitation and potential evapotranspiration are possible indicators of a decrease in humidity in the area. These tendencies are consistent with current projections for aridification in Guatemala (e.g., Pons et al. [37]). Humidity is one of the main determinants of passalid distribution [38]. Thus, the observed upward shift of the species turnover line could be due to the displacement of the montane species by the lowland to mid-elevation species that are more tolerant of less humid conditions.

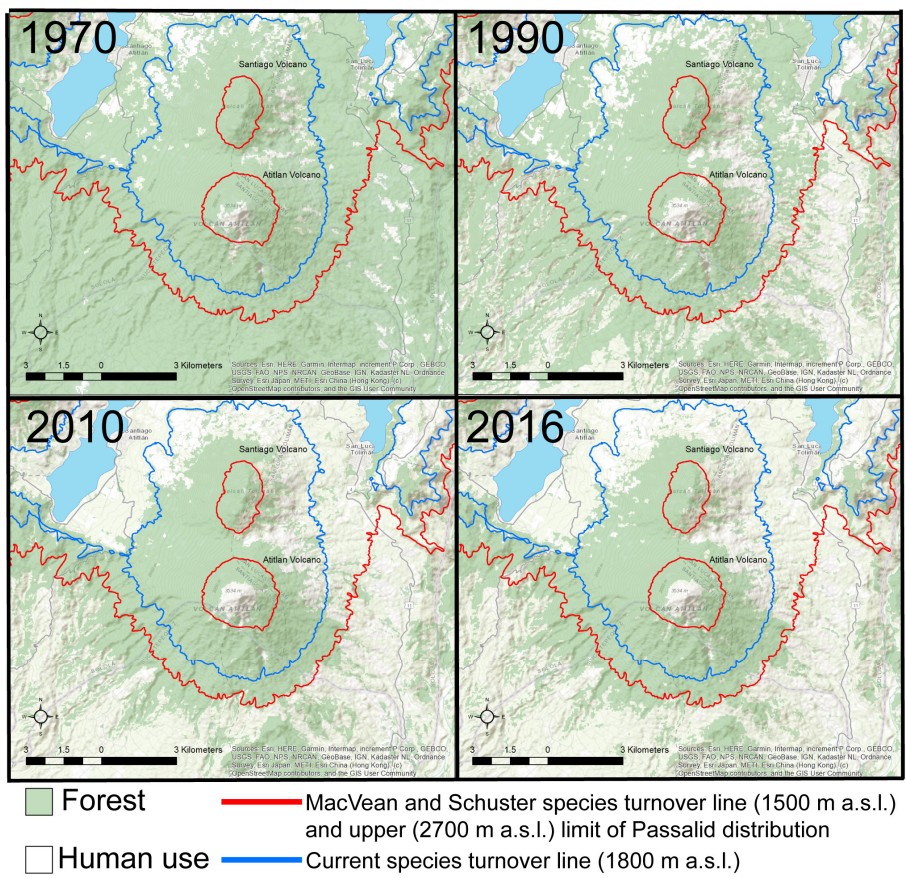

**Figure 4.** Changes in the forest coverage of the Atitlan volcano between 1970 and 2016. Forest coverage data were obtained from [39,40].

(2)   For both volcanoes, passalid beetles were present above 2700 m a.s.l. MacVean and Schuster [9] found passalid beetles between 550 and 2650 m a.s.l. (Figure 3C). They reported sampling 25 logs above 2700 m a.s.l. at Atitlan and Santa Maria, but these logs did not contain passalid colonies. In the current study, logs containing passalid beetles were found from 1043 to 3498 m a.s.l. at Santa Maria, and 850 to 2900 m a.s.l.

at Atitlan (Table 1). The species found above 2700 m a.s.l. were *C. granulifrons*, *Ps. junctistriatus*, and *Proculini* sp. "*incertae sedis*", which are all montane endemic species.

Although MacVean and Schuster [9] did not record passalid beetles above 2700 m a.s.l. at Atitlan and Santa Maria, the authors suggested that passalid beetles were "rare" at higher altitudes and unlikely to be encountered. One main difference between the two volcanoes is the change in land use in the montane forest. Santa Maria has lost more of its original vegetation than Atitlan. Perhaps Santa Maria has fewer available logs at higher altitudes than Atitlan. Though rare, passalid beetles are more likely to be found because they are more concentrated in a scarce resource. This hypothesis, however, has not been rigorously tested.

The three species of passalids found above 2700 m a.s.l. at Santa Maria and Atitlan are all characteristic of high-altitude forests. At least *C. granulifrons* is known to occur above this altitude [9,17] and in other montane regions of Guatemala and Mexico [13,17,18]. *Ps. junctistriatus* is known to occur in the GSVC and Chiapas [13,17,18]. The undescribed species collected has not been recorded outside the volcanic chain (and is therefore likely to be a local endemic) and constitutes the highest altitudinal record of a passalid beetle in nuclear Mesoamerica.

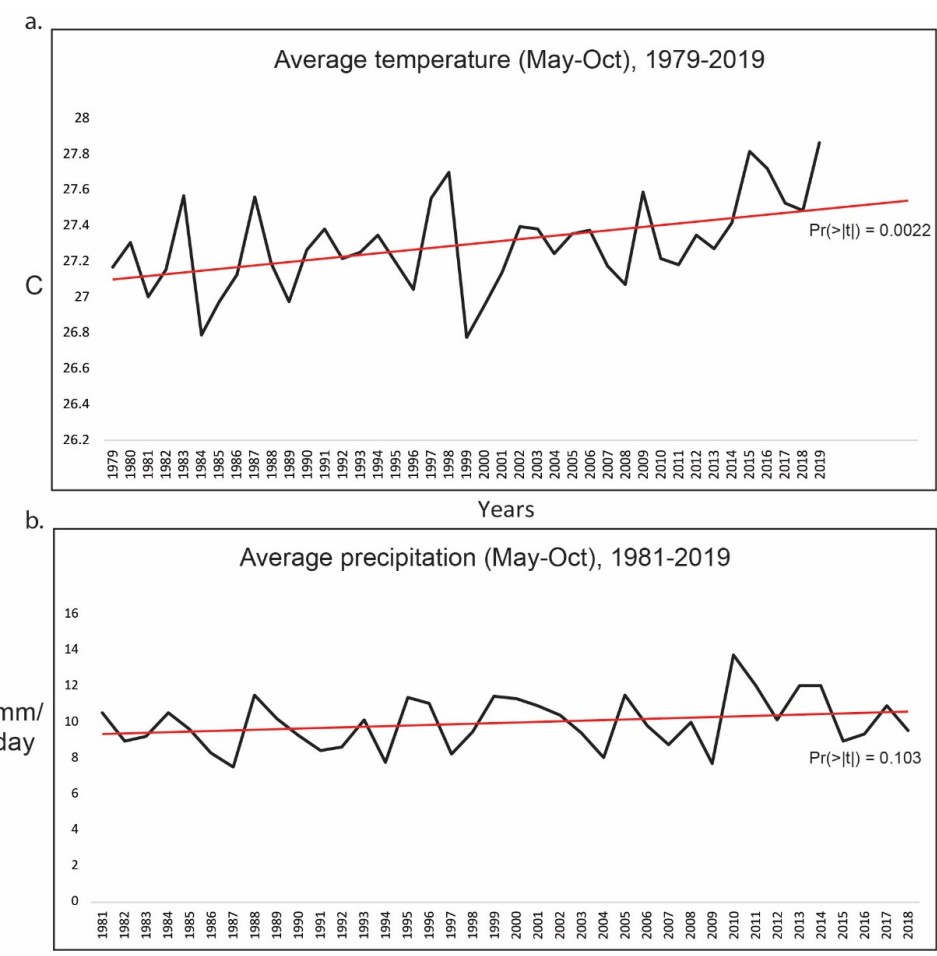

**Figure 5.** *Cont.*

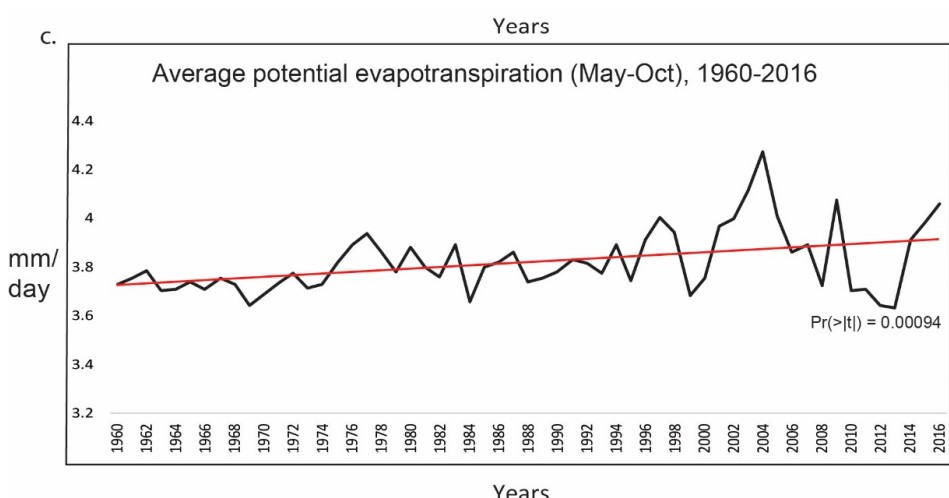

**Figure 5.** (**a**) Historical average temperature for the rainy season (May to October) in the area of interest for the period 1979 to 2019; the temperature is given in degrees C (data obtained from [29]). (**b**) Historical average precipitation for the rainy season (May to October) in the area of interest for the period 1981 to 2018; precipitation is given in mm/day (data obtained from [30]). (**c**) Historical average evapotranspiration for the rainy season (May to October) in the area of interest from 1960 to 2016; evapotranspiration is given in mm/day (data obtained from [29]). The black lines represent the historical data and the red lines represent the linear tendencies. *p* values (Pr(>|t|)) are given for the lineal regression t-test for each model; statistical significance was considered when Pr(>|t|) < 0.05.

These observations are consistent with the shift of the passalid beetle community to higher elevations, likely in response to changes in local climate/habitat conditions, including increased temperatures and changes in forest composition. Species subjected to these range shifts are at risk of mountaintop extinctions, range-shift gaps, and biotic attrition [3,41–43]. Furthermore, these risks are exacerbated when coupled with rapid habitat loss and fragmentation [41]. Given this, humid montane forest sky islands should be considered priority conservation areas. Furthermore, our findings suggest that protection of the forest coverage alone might not be sufficient to counter the effects of climate change on montane species. We need more information to understand the dynamics and causes of these shifts to develop effective conservation strategies.

**Supplementary Materials:** The following supporting information can be downloaded at https://www.mdpi.com/article/10.3390/d15030315/s1, File S1: Geographical coordinates of the collecting points in the current study, with a list of species found at each spot. File S2: Presence–absence matrix for 1982 and current study; File S3: Linear regression and slope significance test for mean temperature, mean precipitation, and mean potential evapotranspiration. Figure S1: Distribution map of the collecting points along Atitlan Volcano; Figure S2: Distribution map of the collecting points along Santa Maria Volcano.

**Author Contributions:** Conceptualization, C.F.B.-B., C.R. and J.C.S.; methodology, C.F.B.-B., C.R. and D.P.; software, C.F.B.-B.; validation, C.F.B.-B., D.M. and J.C.S.; formal analysis, C.F.B.-B.; investigation, C.F.B.-B. and C.R.; resources, D.M.; data curation, C.F.B.-B. and C.R.; writing—original draft preparation, C.F.B.-B.; writing—review and editing, D.M., J.C.S. and D.P.; visualization, C.F.B.-B. and C.R.; supervision, D.M. and J.C.S.; project administration, C.F.B.-B. All authors have read and agreed to the published version of the manuscript.

**Funding:** This research received no external funding.

**Data Availability Statement:** All data from this study are included in the supplementary materials.

**Acknowledgments:** We thank Passalid expert Enio Cano for his assistance with the study. We thank Faustino Camposeco, Julio Amschel López, Miriam Saraccini, Samuel Secaira, and Edi Tocón for their assistance in the field. We also thank Dave Clarke, Jennifer Mandel, Keith Bowers, Matthew

Parris, and Randall Bayer for their input. Finally, we thank Stephanie Haddad, Mary Liz Jameson, Rob Dunn, Seema Seth, and Michelle Kirchner for the manuscript pre-review.

**Conflicts of Interest:** The authors declare no conflict of interest.

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
