# Peer review of "Replicate Studies Separated by 40 Years Reveal Changes in the Altitudinal Stratification of Montane Passalid Beetle Species (Passalidae) in Mesoamerica"

_diversity, doi:10.3390/d15030315_

Round 1
Reviewer 1 Report
Overview:
The authors repeated a beetle survey performed in 1981 on two volcanos in Guatemala. Comparing the two elevational gradients at two points in time, they found more drastic changes in passalid beetle species composition in higher altitudes than in mid elevations. The manuscript presents a simple but interesting study. I suggest to reflect conservational implications at the end of discussion and have some minor comments regarding methods and results sections:
Minor comments:
Line 123: Mention how many individuals you collected in total on both volcanos.
Line 127: Was this the case also in 1981? I fully understand the decision, but wouldn’t you expect changes in beetle presence or absence due to a different habitat formed by the changes in exposition and consequently different forest type or tree species?
Line 131: How many replicates (rotting logs) did you sample at each elevational step?
Line 163: What’s “m”?
Line 182: Describe how you detected those turnover lines.
I would move figure 4 to results section.
Include a brief description (and relocate figure 5) of detected changes in climatic variables in results section.
Since you mention in introduction that you hope to inform conservation efforts in the region, I strongly recommend to close this circle at the end of discussion and to come back to this issue e.g. reflecting on threats or required conservational actions in the region.
Author Response
We want to thank the suggestions made by reviewer 1. Please find our reply to their comments in the attached document

Reviewer 2 Report
This study addresses an important issue: changes in the altitudinal range of arthropods on a multi-decadal timescale in a tropical mountain, as a result of global change. The study area is especially interesting and challenging in this regard, as the effects of deforestation combine with those of climate change. The work is based on a good knowledge of the target beetle family and of the ecological context of Guatemala mountains, an expertise that a priori guarantees the quality of the results. So, such a study is a very welcome contribution to an emerging field of research.
However, I regret to point out several methodological flaws that call into question part of the results.
A minor issue first: The introduction begins with a geographic description of the study site instead of the expected presentation of a concrete scientific objective or a problem to be solved, which should have been followed by a short state of the art, at a broader scale than only Guatemala and the family Passalidae. The growing number of published studies about resurvey of altitudinal transects in tropical mountains, based on insects, has been completely overlooked.
The major issue is the reliability and representativeness of the observations made, as (I quote, l. 276) “Without abundance or probability occurrence data for the species in the area, it is difficult to know if the observed differences are due to ecological changes, substrate availability (rotting logs), or collecting effort”. The following lines reveal, almost at the end of the manuscript, that the sampling protocols best adapted to passalids are still under debate, and that their abundance and diversity still cannot be properly estimated. This fact suddenly calls into question the results presented previously. This issue must have been be addressed from the beginning, and the method section must have reflected an ad hoc strategy to remedy or mitigate it. Just one example: the abstract highlights “a shift of the passalid beetle community to higher elevations”, based on a 859 m difference in the highest point where passalids have been collected at Santa Maria. This would be a enormous upslope shift – if it really is a shift of the species’ top limit, not an artifact of a sampling method providing non-comparable results.
Therefore, if this study aims to “help establish a baseline for the passalid beetle community in the GSVC”, a well-defined sampling method should have been designed. On the contrary, the Material and method section is very vague. A map showing the precise location of the transect stations and a table with their geographical coordinates are missing. Figure 1 does not give this compulsory information. Were the transects exactly the same in 1981 and in 2017? This key information is vaguely suggested: “we attempted to replicate the transects » (l. 124), but does « attempted » mean that the transect were not exactly the same? The way the insects were collected (or simply observed?) is not indicated, nor is the intensity of the collection effort. How many repetitions at each station? This information should be given for the 1981 survey as well as for the more recent one.
L. 145, I don’t understand what “every 200 m” exactly means. Sample stations regularly placed every 200 m in altitude? Or 200 m wide zones in which collections were randomly made? Moreover, the use of the adverb “approximately” is of concern as it increases uncertainty.
L. 137: J. Schuster cannot be one of the authors and, at the same time, be mentioned here as an outside expert.
Author Response
We want to thank reviewer 2 for their suggestions. Please see our reply to their comments in the attached letter.

Round 2
Reviewer 2 Report
This reviewer thanks the authors for the changes made in this new version of the manuscript. All the issues raised have been fully and satisfactorily addressed.